# Creating Opportunities to Eliminate Disparities in Lung Cancer Outcomes: A Call for Diverse Study Populations. Comment on Kohan et al. Disparity and Diversity in NSCLC Imaging and Genomics: Evaluation of a Mature, Multicenter Database. *Cancers* 2023, *15*, 2096

**DOI:** 10.3390/cancers15153762

**Published:** 2023-07-25

**Authors:** Caretia JeLayne Washington, Dejana Braithwaite

**Affiliations:** 1Department of Epidemiology, University of Florida, 2004 Mowry Rd., Gainesville, FL 32610, USA; 2University of Florida Health Cancer Center, University of Florida, 2033 Mowry Rd., Gainesville, FL 32610, USA; 3Department of Surgery, University of Florida, 1600 SW Archer Rd., Gainesville, FL 32608, USA

We read with extensive interest the recently published paper, by Kohan et al., “Disparity and Diversity in NSCLC Imaging and Genomics: Evaluation of a Mature, Multicenter Database” [1]. The authors utilize a novel approach in exploring disparities in non-small-cell lung cancer (NSCLC) by assessing differences in imaging, genetics, and outcomes by race. The primary finding of this study was that the only racial disparity discovered was that the prevalence of 3-month surveillance imaging was higher amongst White patients, which could provide substantial insight into the disparities in NSCLC if the study population was representative of the general population.

Given that disparities in NSCLC is a vital area of research that needs more attention, the purpose of this letter is to question some of the characteristics of the study population and the conclusions drawn about disparities. There are three main areas of concern: (1) the dichotomization of race to explore racial disparities, (2) that ancestry was not examined for its contribution to genetic diversity, and (3) the limited representation of sociodemographic characteristics of the study population.

The authors discussed dichotomizing race into White and non-White due to the small number of non-White participants in the sample; however, grouping all non-White participants in one comparison group overlooks the different disparities that racial minority groups face. For instance, Black people make up 12.4% of the US population [2], yet Black people only made up 5.1% of the study population. Extant literature has shown that Black people develop lung cancer at lower smoking intensities, have lower uptake and adherence to lung cancer screening, are less likely to receive guideline-recommended imaging surveillance following curative-intent therapy for early stage disease, and have higher mortality rates than White patients [3,4,5]. Although the authors collected multi-center data across North America, the racial/ethnic diversity of the study population was limited, as Hispanic ethnicity was not included. To increase the generalizability of research findings, future studies may benefit from expanding cohort consortia, led by the National Cancer Institute, the International Agency for Research on Cancer, and other organizations, by including and engaging diverse patient populations. 

Genomic diversity was also assessed by self-reported race, yet genetic drivers are more related to ancestry than self-reported race [6]. It is important to assess ancestry through genomic and statistical technologies because individuals within the same racial category can have genetic differences due to different ancestral lineages, also known as genetic admixture [7]. For example, the study found that the epidermal growth factor receptor (EGFR) was more frequently mutated amongst non-White patients and was more likely to confer a survival benefit than a TP53 mutation and a KRAS mutation. However, the prevalence of the EGFR does not have the same prevalence within and across racial/ethnic population subgroups, which poses a challenge for estimating the overall survival of NSCLC patients from racial and ethnic minority populations [8]. Additionally, the study did not assess whether racial and ethnic minority populations received equivalent targeted therapy for EGFR-mutated NSCLC, which may also impact prognostic estimates. Crucially, there is a paucity of information in genome-wide association studies regarding polygenic risk score distribution in diverse populations, which remains a key open question for future research [9].

In addition to ethnicity and ancestry, other sociodemographic factors need to be included in the study to evaluate disparities in NSCLC imaging and outcomes, such as social determinants of health, including but not limited to socioeconomic status and geographic location (e.g., rurality), and clinical factors, such as comorbidity burden. Notably, a study from the National Cancer Database found that NSCLC patients residing in rural areas were at an increased risk of death versus those residing in urban settings [10]. Furthermore, racial and ethnic minority groups are disproportionately affected by comorbidities, and comorbidity burden can negatively affect NSCLC prognosis [11,12,13]. This analysis underscores the importance of considering multi-level clinical and sociodemographic factors that contribute to these inequities.

Despite generalizability-related limitations, this study highlights the importance of examining understudied aspects of clinical lung cancer research, such as imaging and genetic diversity, and how they may contribute to racial/ethnic disparities in NSCLC outcomes. Whenever possible, future research should evaluate the main effects of race/ethnicity, assess ancestry informative markers, and incorporate relevant sociodemographic and clinical factors into analyses to comprehensively examine disparities in NSCLC outcomes.

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
