# Peer review of "Creating Opportunities to Eliminate Disparities in Lung Cancer Outcomes: A Call for Diverse Study Populations. Comment on Kohan et al. Disparity and Diversity in NSCLC Imaging and Genomics: Evaluation of a Mature, Multicenter Database. Cancers 2023, 15, 2096"

_cancers, 2023, doi:10.3390/cancers15153762_

Round 1

Reviewer 1 Report

Review comments for “Creating opportunities to eliminate disparities in lung cancer outcomes: a call for diverse study populations.”

Summary:

This letter introduced the research of Kohan et al which examined racial disparities in non-small cell lung cancer (NSCLC) by analyzing imaging, genetics, and outcomes. They found a disparity in surveillance imaging, with higher rates among White patients. However, the authors pointed out the concerns about the study. 1. population, as it grouped all non-White participants together and lacked representation of Hispanic ethnicity. 2. Genetic diversity based on self-reported race was assessed, but ancestry is more relevant for genetic drivers. 3. Sociodemographic factors like socioeconomic status and comorbidity were not fully considered.

Future research should evaluate specific racial/ethnic groups, incorporate ancestry analysis, and include a comprehensive range of sociodemographic factors to better understand NSCLC disparities.

Comment:

This is a great letter. The authors not only introduced a novel approach from the research of Kohan et al. but also pointed out the drawback and the limitation of the study. In the meantime, the authors provided the suggestions for the future research in the field which may contribute to the racial disparities in NSCLC outcomes.

Author Response

We thank you for your positive feedback.

Reviewer 2 Report

1.This research focused on Creating opportunities to eliminate disparities in lung cancer outcomes: a call for diverse study populations ,this was a letter type manusrcipt, comments about the article“Disparity and Diversity in NSCLC Imaging and Genomics: Evaluation of a Mature Multicenter Database.

2.The authors have pointed out the shortcomings and limitations of this article, and also confirmed the positive aspects, the entire comment is well founded and well completed.

3.But It is very one-sided to only discuss white and non white people, because Yellow people make up a large proportion.

4.There are so many genes mutation related with NSCLC, but only discuss the main such as EGFR and KRAS.

5. How to do the diverse study populations?Collaborating with Chinese lung cancer researchers? Can you give some plan?

Author Response

Thank you for your comprehensive review of the manuscript.

5.) We think that this is an excellent suggestion. We have incorporated a statement in each paragraph that suggests a strategy or plan to address the issue that was raised.

Reviewer 3 Report

The authors present a comprehensible argument. I agree that racial differences can be categorized not only between whites and non-whites but also among non-whites, including Asians and non-Asians. In my opinion, it is necessary for Dr. Kohan and his colleagues to address the questions raised by the authors.

Author Response

Thank you for reviewing and emphasizing the importance of addressing these questions.